# Emerging Yeast Infections in Confined Elderly: Niche Effects and Rapid, Opaque Diversities Affecting Multisciences for Strains of *Candida*/*Clavispora*/*Candidozyma* Complex

**DOI:** 10.3390/microorganisms13092007

**Published:** 2025-08-28

**Authors:** Donald G. Ahearn

**Affiliations:** Microbiology (Emeritus) Center for Applied & Environmental Microbiology, Georgia State University, Atlanta, GA 30303, USA; dahearn@gsu.edu

**Keywords:** *C. haemulonii* Complex, rapid phenotypic–genotypic diversities, intrinsic drug resistance, confined elderly, sRNA-mitDNA epigenetic functions, yeast gold standards

## Abstract

Undetected, rare diversities scattered among medically important emerging yeast, both in historical literature and recent AI research, are questioned and proffered for rapid and variable niche adaptations. These variable stresses may appear most notable among the medical sciences concerned with emerging yeast infections. Misidentified species, questionable therapies, and possible disease functions and retractions or rejections of manuscripts are identified in part, with the *C. haemulonii* Complex, *C. auris*, and *C. lusitaniae*. Analysis of the Commentary below, including a thorough definition of the disputed details among the references, implies that further researches on EVs and sRNA are needed by multi-sciences.

## 1. Introduction

Clonal instabilities and environmental stresses may underlie the variance in the taxonomic species–strain distinctions, geographical distributions, commercial applications and treatment modes for emerging yeast infections involving the *Candida/Clavispora/Candidozyma* Complex(es). Obscure stresses ranging from exposure to blood sepsis in the confined elderly, adaptive histories in culture collections, marine coastal sites and laboratory processes may interact with a rare strain or progeny of the *C. haemulonii* Complex, *C. auris* (a supposed killer yeast co-infecting with SARS-2-COVID-19), *C. lusitaniae* or similar emerging new species considered health threats [1,2,3,4,5,6,7,8]. Cryptic unstable PCR target sites with potential for rapid adaption to environmental stresses appear to be strain-related. Niche multi-stresses; sequences and intensities; and the epigenetic, rapidly adaptive and possibly dimensional involvement of sRNA-DNA may alter functions among given strains. Misidentifications, insufficient genomic barcodes, strain-related virulence, drug resistance and transient commensal functions may remain flexible or bound to a given culture due to the status of the technologies in a laboratory and ecological investigations [1,3,8,9,10,11]. In 2025, SARS-free candidemia infections by multiple-drug-resistant species are reportedly increasing among the compromised and confined. This SARS-free or COVID-19 terminology essentially applies in the USA to the general geographical recognition of coronavirus activities in wastewaters and the extent of varied flu-like symptoms among an immediate population. A specific disease–virulence association for species–strains across the *C. haemulonii* Complex and *C. auris* remains problematic. The compiled literature on recent omics sciences indicates the questionable premature application of a gold-standard categorization to rare clusters of yeast from diverse taxa, e.g., the *Candida parapsilosis* Complex and *Metschnikowia pulcherrima* [7,8,9,10,11,12,13,14]. Such possibilities of genetically unique strain clusters are propagated for the *Candida haemulonii* Complex, for new species considered health threats, for medical staff in routine clinical laboratories, for visiting nurse practitioners servicing the confined elderly and for remote patient monitoring. The treatment of *C. auris* in non-critical patients by various clinicians worldwide now involves an assessment of the patient’s history and the treatment of the syndrome prior to the selection of an antifungal.

## 2. The Status of the Art

The above medical concerns and omnisciences involving industry, taxonomy, medicine and ecology demand further investigation into rapid intrinsic adaptions among opaque clusters of the *Candida*, *Clavispora*, *Metschnikowia* and/or *Candidozyma* Complexes. Whether the current identification practices apply to all strains of the Complex, their potential virulence associations or the binding or locking of unstable PCR target sites by niche stresses remains questionable.

(1)Could old-type and recent cultures of *C. duobushaemolunii*, *C. auris* and *C. lusitaniae* from blood sepsis patients or marine environments, upon sustainment under multi-stresses, e.g., in the neutrophils and macrophages for several generations, provide progeny that gain or lose heat tolerance or amphotericin B and/or azole resistance?(2)Could an evolutionary gene, a dehydration intolerance, rapid sRNA or mDNA epigenetic systems coexist in different combinations and degrees among strains of the Complex?(3)Could recent individual isolates, clonal cells of the *C. haemulonii* Complex (particularly *C. duobushaemolunii* and *C. vulturna*) and their progeny be adapted to grow at 40–42 °C or higher?(4)Could the *C. haemulonii* Complex and *Candida lusitaniae* share common marine ancestors involving possible diverse branching and hybrid relationships?(5)Could an old and a recent strain be interlocked into a micro-niche with an antifungal microbe such as *Streptomyces nodosus*, a marine coral, fish ectoparasites, an apple or a vineyard and subsequently have less fitness to survive in a new niche?(6)Could multi-stresses involving temperatures and niche exposures alter the dimensional arrangements and functions of sRNA or mitRNA-DNA among clusters with genetically diverse domains?(7)Could asci-ascospores evolve or reoccur among haploid clades or subgroups as a secondary feature in a haploid or an aneuploid existence?(8)Could simple repeated testing of a single clonal strain and its progeny (an evolutionary short-term study) provide evidence of cryptic unstable PCR target sites such as ITS, D1/D2 or 26sRNA?(9)Could single domains (“drug-resistant” or “mating type”) in a clonal culture (strain) under niche stress become functional or nonsense in diverse metabolic or genomic pathways?(10)Could a hidden strain of *C. auris* have the potential for a grave co-infection with another yeast strain or virus among various elderly? Does long COVID remain an issue?

The majority of the disease–virulence associations in the literature for the above taxa have been inferred using pre-identified cultures sourced from hospitals and culture collections. Some early concatenated barcodes appear to be insufficient for defining specific strain clusters. Unrecognized niche effects could block or reorient genomic sites, affecting functions distinct from those in *C. albicans* to varied degrees. Also, species–strains of the Complex and similar species may be depicted with unstable functions (genotypic and phenotypic) and surmised to be additional health threats. Future investigations with a focus on the possible diversity of the virulence of unstable domains among such clonal strains under multi-stresses seem essential. Presently, of immediate importance, treatment of the syndrome in the confined elderly prior to the application of currently recognized antifungals such as amphotericin B, azoles and echinocandins appears to be advisable for the rapidly adaptive *Candida/Clavispora*/*Candidozyma* Complexes.

Candidiasis and candidemia caused by non-epigenetic *C. albicans*, *C. parapsilosis* or *C. tropicalis*, with potential drug resistance, more commonly identified among the elderly, require continued updated medical direction (MD) with emphasis on the host syndrome. Annual antiviral therapy for protection against possible co-infections should be maintained. The high past mortality rates estimated for patients linked to SARS and bloodstream infections with *C. auris* appear in stark contrast to the current low mortality rates among the uncompromised elderly. The compiled literature suggests the involvement of loss of fitness of rare, opaque epigenetic functions rather than chromosomal dominance that could more often involve yeast and patient interactions. The virulence factors implied for *C. auris* due to studies in healthy and colonized–diseased hosts (particularly redundant AI reviews and research using insufficient data on mortalities and treatments) may need to be re-addressed for today’s AI projections of strain diversities. The status in time for the diversities in any niche appears foremost to be persistent among these unique Complexes. These considerations and conjectures based on the current clonal progeny may apply to other rare genetic fungal clusters [1,2,3,4,5,6,7,8,9,10,11,12,13,14,15,16]. Some assumed health threats might be misleading; certain strains could continue as a beneficial cohort for humans rather than a rare disease threat.

Yeasts identified with *Torulopsis haemulonii* isolated in the late 1950s due to the conundrum of *Candidozyma* may have been misidentified—or not detected—using past or current AI technologies and their algorithms. Opaque rapid alterations among their old clonal cultures and their progeny may be deduced or observed from the recent AI-compiled literature involving *Candida/Clavispora* and *Candida lusitaniae*. Clusters of strains among various species have unstable target sites that may have led to misleading interpretations, potentially contradicting or harming the multisciences. The rapid development of clonal cultures on agar plates or broths has been noted, but disease functions are often misinterpreted with comparisons to other *Candida* spp.

## 3. Discussion and Further Research

*Candida auris*, first mentioned in 2007, has been misidentified in multiple literature works, most likely as *Candida duobushaemulonii* and the *C. haemulonii* Complex [1]. These misidentifications are common and remain today because of specific PCR rapid systems, particularly for the verification of taxa. During the SARS-2-COVID-19 viral pandemic peak (2019–2022), *C. auris* was considered a killer yeast, particularly for the confined elderly. This stigma remains today, but it mostly concerns those who are critically health-compromised. The morbidities associated with common flu viruses during this period appear to be ill defined as pertaining to sRNA, a potential key for virulence functions [7].

During 2023–2025, the recognition of *C. auris* increased, while associated mortalities decreased significantly. Advancements in technology and the economics of multi-science communications have clouded the current data further, up to today. AI, an exciting gem now and for the future, appears to be challenged by resolving the disease functions for rare unstable fungi, notably among yeasts such as the *C. haemulonii* Complex. Currently, clinical cases and swab isolations of *C. auris* seem to have decreased in hospitals and assisted living facilities. Some of these reports have examined only a few new isolates in comparison with multiple strains and their data from the past literature.

Are grave infections reported for premature babies and the compromised elderly, particularly males with blood sepsis, caused by diverse strains among the *Candida haemulonii* Complex? Are transient skin commensals of environmental niches, particularly in the toe webs, axilla and groin, misidentified as stresses of the flu or *C. auris*? The mortality rates for *Candidozyma*, including the Complex and *C. albicans*, have decreased significantly from the estimates for SARS hospitalization peaks from 30–60% to less than 3–20% for critical patients with or without candidemia. Does this statement reflect the uncommon recognition of early strains or the accumulation of rare isolates at special centralized centers? Are these cultures mostly of tropical or subtropical origin? Could the migration of birds, insects, marine life and humans establish or be involved in a transient, geophysical niche? Are isolates resistant to one or two drugs, such as azoles or amphotericin B, more virulent than pan-resistant strains? Further research on these topics is required.

To address the above, the state of the art in science/technology must be considered with details of any variations in the prior methods, including the sources and identification numbers of clonal types from varied culture collections. Specific species or strains should be labeled with a scientific acronym for naming convention purposes. Terminologies for the distinction of functions for hosts and infectious fungi need to be incorporated for future and more accurate AI interpretation.

Articles below available online as of 13 September 2024. References selected and limited to 16 of over 100 related reviews and research investigations historically compiled/examined without personal laboratory manipulations of clonal Type cultures of *C. auris* after 2014. Aspersions or direct quotes from published or revised manuscripts have been avoided.

## Data Availability

No new data were created or analyzed in this study. Data sharing is not applicable to this article.

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
