# Peer review of "Emerging Yeast Infections in Confined Elderly: Niche Effects and Rapid, Opaque Diversities Affecting Multisciences for Strains of Candida/Clavispora/Candidozyma Complex"

_microorganisms, 2025, doi:10.3390/microorganisms13092007_

Round 1
Reviewer 1 Report (Previous Reviewer 2)
Comments and Suggestions for Authors
This "revised" manuscript has not been revised according to both reviewers and therefore I can't further review this. My previous decision stands.
Author Response
Comments:
This "revised" manuscript has not been revised according to both reviewers and therefore I can't further review this. My previous decision stands.
Author's Reply:
This commentary is designed to ask questions and offer innovative avenues of research for intrinsically adaptive yeast clusters. This brief commentary with its selected references is directed towards the taxonomy, medical interests, various industries and ecologies, and developing AI's on these topics that may change, even daily. For example, doing Google AI Overview search results (pending subject) can change over a short period of time, or the previously found results, sites or document links may no longer be available.
The specific taxonomy in the title is used for direction to diversities among Candida/Clavispora and C. duobushaemulonii among their most recognized associations. This is all meant to be the forerunner for probable future research by other experts in the field of rare diversities. The fragmented nature of my piece is not meant to be necessarily sequential, per se, but the information contained within is directed towards the multi-sciences and to garner further discussion in these matters.
Reviewer 2 Report (Previous Reviewer 1)
Comments and Suggestions for Authors
The revised manuscript shows improvement, and this reviewer have no further questions.
Author Response
Comments:
The revised manuscript shows improvement, and this reviewer have no further questions.
Author’s Reply:
The brief changes in this minor revision appears in accordance with most of the current reviewers (3 out of 4, or 4 out of 5?)
Reviewer 3 Report (New Reviewer)
Comments and Suggestions for Authors
In my humble opinion, this article is very interesting to read, to reflect on the aspects raised and to discuss among scientists dealing with potentially pathogenic ascomycete yeasts, their distribution in the environment, their adaptation to temperatures above 38 degrees, the activation and intensity of the pathogenic properties of the strains. The existing gap between the data collected on clinical strains (more data) and on natural isolates (less data) influences to some extent the bias in the use of AI. Perhaps it would be interesting to add some options for desirable directions of future research to collect information on strains (environmental substrate monitoring studies in different regions, rapid assessment of virulence (in vitro or/and in vivo using host insect models, etc.) of isolated strains of potential pathogens and their temperature growth limits. And for clinical isolates, it would be particularly interesting to study strains isolated from children and adults who have been exposed to stressful conditions over a long period of time that may have compromised their immunity (e.g. those who have lived in war conditions for a long time, etc.).
Author Response
Comments:
In my humble opinion, this article is very interesting to read, to reflect on the aspects raised and to discuss among scientists dealing with potentially pathogenic ascomycete yeasts, their distribution in the environment, their adaptation to temperatures above 38 degrees, the activation and intensity of the pathogenic properties of the strains. The existing gap between the data collected on clinical strains (more data) and on natural isolates (less data) influences to some extent the bias in the use of AI. Perhaps it would be interesting to add some options for desirable directions of future research to collect information on strains (environmental substrate monitoring studies in different regions, rapid assessment of virulence (in vitro or/and in vivo using host insect models, etc.) of isolated strains of potential pathogens and their temperature growth limits. And for clinical isolates, it would be particularly interesting to study strains isolated from children and adults who have been exposed to stressful conditions over a long period of time that may have compromised their immunity (e.g. those who have lived in war conditions for a long time, etc.).
Reply:
I agree with your comments. Hence, the directions, questions, and references provided in my commentary. Your statements allude to my experience with patients I have come across in my field of study. Regarding AI, my sentences have been modified as to your review.
Reviewer 4 Report (New Reviewer)
Comments and Suggestions for Authors
The commentary presents several important questions and demonstrates great engagement with the topic. The issues discussed in the manuscript are highly relevant and noteworthy, however, the text lacks clarity and coherence, necessitating substantial structural revision to enhance readability and conceptual support. It is recommended that the title incorporate the full taxonomic names of the species discussed. Moreover, several sentences require reorganization, as their current syntax frequently impedes clarity and hinders the reader’s ability to follow the author's line of reasoning.
Furthermore, the first part of the manuscript text should provide a concise and clear overview of the topic, delineate the central issues, and explicitly state the hypotheses under consideration. In the present version of the manuscript, the narrative is fragmented, with thematic elements interwoven and abbreviations or explanatory details introduced non-sequentially, thereby disrupting logical progression and manuscript flow, especially the second part of the text. For example, in line 100, the terminology relating to AI is introduced, the acronym itself is explained much later (line 125), including other related issues. What is the validity of these problems relating to AI should be explained earlier and more clearly.
Author Response
Comments:
The commentary presents several important questions and demonstrates great engagement with the topic. The issues discussed in the manuscript are highly relevant and noteworthy, however, the text lacks clarity and coherence, necessitating substantial structural revision to enhance readability and conceptual support. It is recommended that the title incorporate the full taxonomic names of the species discussed. Moreover, several sentences require reorganization, as their current syntax frequently impedes clarity and hinders the reader’s ability to follow the author's line of reasoning.
Furthermore, the first part of the manuscript text should provide a concise and clear overview of the topic, delineate the central issues, and explicitly state the hypotheses under consideration. In the present version of the manuscript, the narrative is fragmented, with thematic elements interwoven and abbreviations or explanatory details introduced non-sequentially, thereby disrupting logical progression and manuscript flow, especially the second part of the text. For example, in line 100, the terminology relating to AI is introduced, the acronym itself is explained much later (line 125), including other related issues. What is the validity of these problems relating to AI should be explained earlier and more clearly.
Author's reply:
This commentary is designed to ask questions and offer innovative avenues of research for intrinsically adaptive yeast clusters. This brief commentary with its selected references is directed towards the taxonomy, medical interests, various industries and ecologies, and developing AI's on these topics that may change, even daily. For example, doing Google AI Overview search results (pending subject) can change over a short period of time, or the previously found results, sites or document links may no longer be available.
The specific taxonomy in the title is used for direction to diversities among Candida/Clavispora and C. duobushaemulonii among their most recognized associations. This is all meant to be the forerunner for probable future research by other experts in the field of rare diversities. The fragmented nature of my piece is not meant to be necessarily sequential, per se, but the information contained within is directed towards the multi-sciences and to garner further discussion in these matters.
Round 2
Reviewer 4 Report (New Reviewer)
Comments and Suggestions for Authors
The author’s explanations are acknowledged.
Author Response
Comments to Reviewer #4:
Your first review was to reject, and format as a structured research article. My terminology was not understandable at that time. My reply indicated that you did not understand that this was a commentary, or a letter to the editors, and that your overall experience with yeasts appeared limited. My revision (as of 8/16/2025) addressed your concerns as well as the comments of the other editors. Your reply to the previous revision was to not accept my changes because you were "too busy". This is a problem that I can well understand. Feel free to provide anonymous comments in the future.
This manuscript is a resubmission of an earlier submission. The following is a list of the peer review reports and author responses from that submission.
Round 1
Reviewer 1 Report
Comments and Suggestions for Authors
This perspective article explores the Candida haemulonii Complex, Candida auris, and related species, emphasizing clonal instability, phenotypic and genotypic diversity, and the persistent challenges in species classification, virulence characterization, and therapeutic decision-making. It brings attention to the ecological, clinical, and diagnostic complexities of these emerging fungal pathogens and situates the discussion within the broader context of rising drug resistance, particularly in vulnerable, aging populations and post-COVID healthcare settings. The manuscript presents a series of focused research questions addressing the roles of environmental stressors, possible marine ancestry, and epigenetic mechanisms involving sRNA and mDNA, which could inform future investigations. By drawing from ecological, medical, taxonomic, and technological perspectives—including concerns about misidentification through AI-based tools—it highlights the multifaceted nature of these organisms. Importantly, the article challenges the early labeling of certain strains as “killer yeasts,” cautioning against overdependence on incomplete genetic barcodes or conclusions drawn from limited isolate studies.
Comments:
The manuscript would benefit from a more clearly defined structure, with distinct sections such as Background, Hypotheses, and Future Directions. Many sentences are densely written and highly speculative, making it difficult for readers to follow the flow of ideas due to limited transitions and underdeveloped arguments. While the numbered list of hypotheses (lines 45–72) is a useful organizational feature, each item would be more impactful with additional explanation and contextual detail.
The introduction would benefit from a clearer articulation of the article’s objectives—specifically, what central questions or aims it seeks to explore. To enhance clarity and flow, the hypotheses could be reorganized into thematic categories such as drug resistance, ecological origins, and diagnostic limitations. Additionally, providing concise summaries or key takeaways after the list of questions would help orient readers toward potential interpretations, solutions, or future experimental approaches.
Although many ideas presented are thought-provoking, several claims—such as the role of marine ancestors or the notion of fungal commensals as "beneficial cohorts"—would benefit from stronger empirical support. Statements like “cryptic unstable PCR target sites...appear to be strain related” should be clarified by specifying the exact loci involved and the nature of their instability. The manuscript uses several ambiguous or unconventional terms (e.g., “Gold Standard categorization,” “dimensional involvement of sRNA-DNA,” “status of the arts”) without adequate explanation.
Including figures to visually depict clade relationships or phenotypic shifts under stress conditions would greatly enhance clarity and support key points in the manuscript.
Reviewer 2 Report
Comments and Suggestions for Authors
The commentary Considerations: Clonal instabilities and niche effects among the emerging yeast infections involving the Candida/Clavispora Complex; are strain clusters among clades of C. auris intermingled with C. duobushaemulonii? is a string of random thoughts and notes that are currently unreadable. It seems there is no introduction to the topic or these organisms. What is actually commented on? There is little reference to primary literature and what is actually the point of this written piece. It reads terribly, and the questions asked seem to be relatively random as well.
It would urge the author to structure their thoughts and read up on literature on Candida auris and haemulonii.
Comments on the Quality of English LanguageThe wording of most sentences is incredibly complex which makes this difficult to read, besides all the points mentioned above.
